# Immobilization of Polyethyleneimine (PEI) on Flat Surfaces and Nanoparticles Affects Its Ability to Disrupt Bacterial Membranes

**DOI:** 10.3390/microorganisms9102176

**Published:** 2021-10-19

**Authors:** Nesha May Octavio Andoy, Meera Patel, Ching Lam Jane Lui, Ruby May Arana Sullan

**Affiliations:** 1Department of Physical and Environmental Sciences, University of Toronto Scarborough, 1065 Military Trail, Toronto, ON M1C 1A4, Canada; nesha.andoy@utoronto.ca (N.M.O.A.); meera.patel@mail.utoronto.ca (M.P.); jane310@connect.hku.hk (C.L.J.L.); 2Department of Chemistry, University of Toronto, 80 St. George St., Toronto, ON M5S 3H6, Canada

**Keywords:** polyethyleneimine, PEI, bacterial outer membrane, polydopamine nanoparticles, atomic force microscopy, AFM, quantitative imaging, QI, Young’s modulus

## Abstract

Interactions between a widely used polycationic polymer, polyethyleneimine (PEI), and a Gram-negative bacteria, *E. coli*, are investigated using atomic force microscopy (AFM) quantitative imaging. The effect of PEI, a known membrane permeabilizer, is characterized by probing both the structure and elasticity of the bacterial cell envelope. At low concentrations, PEI induced nanoscale membrane perturbations all over the bacterial surface. Despite these structural changes, no change in cellular mechanics (Young’s modulus) was detected and the growth of *E. coli* is barely affected. However, at high PEI concentrations, dramatic changes in both structure and cell mechanics are observed. When immobilized on a flat surface, the ability of PEI to alter the membrane structure and reduce bacterial elasticity is diminished. We further probe this immobilization-induced effect by covalently attaching the polymer to the surface of polydopamine nanoparticles (PDNP). The nanoparticle-immobilized PEI (PDNP-PEI), though not able to induce major structural changes on the outer membrane of *E. coli* (in contrast to the flat surface), was able to bind to and reduce the Young’s modulus of the bacteria. Taken together, our data demonstrate that the state of polycationic polymers, whether bound or free—which greatly dictates their overall configuration—plays a major role on how they interact with and disrupt bacterial membranes.

## 1. Introduction

The ability of cationic polymers to form membrane defects and holes on both supported lipid bilayer and cell membranes have been widely studied [1,2,3,4]. Due to their membrane destabilizing property and ability to form stable structures with negatively charged nucleic acids, polycationic polymers have been largely investigated as nucleic acids delivery systems for gene therapy applications [5,6]. Among the most commonly used macromolecule for these types of applications is the synthetic, water soluble, and commercially available polycationic polymer, poly(ethyleneimine) or PEI [7]. Owing to the presence of a significant number of primary, secondary, and tertiary amino groups along its chain, PEI exhibits a high cationic charge density, and is very effective in inducing nanoscale defects in both model and cellular membranes [2,8]. Moreover, PEI has been extensively used to decorate other drug nanocarriers, in order to improve their ability to deliver cargoes inside cells [6,9].

As it is well established that some cationic polymers have antimicrobial properties (e.g., antimicrobial peptides), the effect of PEI activity in destabilizing the bacterial membrane has also been studied. It was previously reported that the bactericidal action of PEI is heavily influenced by its size (low vs. high molecular weight) and form (linear vs. branched) [10]. While the low molecular weight-linear PEI is effective, especially against Gram-positive bacteria, the high molecular weight-branched PEI (hMW-bPEI) has been reported to be non-bactericidal [10,11].

Due to the non-biocidal nature of hMW-bPEI, chemical modifications of the amino groups in the polymer have been employed to endow biocidal activity [12]. For example, quaternary amine PEI (QA-PEI) has been synthesized, by alkylation of its amino groups, and has been used to decorate surfaces to act as biocidal agents. QA-PEI has been shown to be active against both Gram-positive and Gram-negative bacteria, fungi, as well as viruses [13,14,15]. Moreover, the non-bactericidal nature of hMW-bPEI has found applications toward bacterial immobilization procedures, which require that bacteria remain viable after surface attachment. PEI has been widely used to firmly attach bacteria to surfaces, especially for scanning probe microscopy experiments [16,17,18]. These prior works have shown that bacteria immobilized on PEI-coated surfaces are very stable, adhering firmly in place and allowing a nano-sized cantilever to map its surface topography with very high spatial resolution—all under physiological conditions [9,17,19]. Moreover, PEI has been used to immobilize bacteria on micron-sized probes for force spectroscopy measurements, to quantify the extent of bacteria-surface interactions [20]. It is worth noting that the PEI-immobilized bacteria remained viable and did not show any structural changes of the outer membrane. However, the hMW-bPEI used to immobilize these bacteria has also been shown to be an effective membrane permeabilizer, causing extensive structural changes on the outer membrane of Gram-negative bacteria [11,21]. The ability of hMW-bPEI to disrupt bacterial membranes has been reported to enhance the action of some antibiotics against Gram-negative bacteria [21,22,23]. In addition, it has been suggested that for bacteria immobilized on PEI-coated substrates, polymers that detach from the surface might cause denaturation of bacteria [17].

Here, we show that while the binding of a 60 kDa (M_n_) hMW-bPEI—a commercially available polymer commonly used for bacterial immobilization— to Gram-negative bacteria, *E. coli*, can induce outer membrane structural changes and alter bacterial cellular mechanics, its ability to alter and destabilize the cellular envelope of *E. coli* is diminished once immobilized. Furthermore, we show that PEI immobilized on a nanoparticle surface interacts in a different manner with bacterial membranes than when immobilized on a flat surface. Our work highlights the major role that surface immobilization plays on how polycationic polymers can interact with and change the physical properties of target membranes.

## 2. Experimental

**Polydopamine nanoparticle synthesis and functionalization with PEI**. Polydopamine nanoparticles (PDNP) were synthesized by dissolving dopamine (Thermo Scientific, Waltham, MA, USA) in a 10 mM Tris buffer pH 10 to make a 0.5 mg/mL solution. Polymerization of dopamine under ambient O_2_ condition was allowed for 24 h at RT with mixing at 60 rpm. The nanoparticles were then harvested via centrifugation (Centrifuge 5804 R, Eppendorf) at 15,000 *g* for 30 min, washed 3× with a 10 mM Bicine buffer pH 8.5 (Sigma-Aldrich, Oakville, ON, Canada), and resuspended and stored in the same buffer. To determine the concentration of nanoparticles (NP) in the solution, harvested NP were washed with nanopure water, then freeze dried. Thereafter, the dried NP were weighed to determine the total mass of nanoparticles in the stock solution. For NP functionalized with PEI (PDNP-PEI), 1 mg/mL of NP solution was mixed with PEI (10 mg/mL, *M_n_* = 60,000, Sigma-Aldrich, P3143) in the presence of 0.1 M KCl (Sigma-Aldrich) in 10 mM Bicine pH 8.5 and incubated for 12 h at RT with constant mixing. The PDNP-PEI was harvested by centrifugation at 20,000 g for 10 min and washed 3× with the Bicine buffer. The final PDNP-PEI was resuspended and stored in 10 mM Bicine pH 7.5. To monitor if functionalization is successful, the hydrodynamic size and the surface zeta potential of bare and PEI-functionalized nanoparticles were determined in 10 mM Bicine pH (7.5) via Dynamic Light Scattering (DLS) and Phase Analysis Light Scattering (PALS), respectively (NanoBrook Omni, Brookhaven Instruments, Holtsville, NY, USA).

**Effect of PEI on bacterial growth**: The *E. coli* AR3110 strain used in this work is a gift from Regina Hengge of the Institut fur Biologie/Mikrobiologie, Humboldt-Universitat zu Berlin. An overnight liquid culture of *E. coli* AR3110 in LB was harvested via centrifugation at 2000× *g* for 5 min, then washed once with PBS. The pellet was resuspended in PBS, then mixed with different concentrations of PEI with final *E. coli* OD 0.1 and incubated for 15 min. Thereafter, 5 μL of each suspension was added to 195 μL M9 minimal media or LB in a 96-well plate. The growth of *E. coli* was then monitored for 20 h at 37 °C at 244 rpm using a microplate reader (Infinite M Nano, Tecan, Männedorf, Switzerland).

**AFM characterization**: An overnight liquid culture of *E. coli* AR3110 in LB was harvested via centrifugation at 2000× *g* for 5 min, then washed once with PBS. The pellet was resuspended in PBS, then mixed with different concentrations of PEI with final *E. coli* OD 0.1 and incubated for 15 min. After incubation, the suspension was filtered using the PEI-coated polycarbonate membrane (PC, 25 mm, 0.1 μm, Millipore, Oakville, ON, Canada). This PC-PEI membrane was prepared by incubating the PC membrane with a 1% PEI solution in water overnight at 23 °C with shaking at 60 rpm. Then, PEI-coated PC (PC-PEI) membranes were washed extensively with nanopure water before bacteria were immobilized. Non-coated PC membranes were used for *E. coli* mixed with 100 μg/mL PEI and PDNP-PEI. PC-immobilized *E. coli* were characterized using the Quantitative Imaging (QI) modality of the Atomic Force Microscope (AFM, Nanowizard 4, JPK Instruments, Berlin, Germany) using Silicon Nitride Probes (SNL-A Bruker). QI force-distance curves were recorded with a relative force setpoint of 2 nN, a *z*-range of 2000 nm, and a (vertical) cantilever speed of 125 µm/s. Prior to use, the spring constant of each cantilever was individually calibrated using the thermal noise method and were found to be in the range of *k* = 352–412 mN/m [24]. Unless otherwise stated, measurements were performed at ~25 °C using M9 minimal media as the imaging solution. Hertz fitting was done on the extended part of the force curve using a parabolic model for the tip with a radius of 2 nm, the nominal radius of SNL cantilevers. AFM data were analyzed using a software that was previously developed in the lab, where we use a selection tool to specifically define areas within the QI map to more accurately extract Young’s modulus values, i.e., only the top middle portion of the bacteria were included in the analysis [25,26].

**Confocal microscopy imaging**: Confocal microscopy (Zeiss LSM510 META, Jena, Germany) was performed to image *E. coli* immobilized on PEI-coated glass coverslips. Glass coverslips were first sonicated in a saturated solution of Alconox detergent for 10 min, then washed with a copious amount of nanopure water with sonication. Thereafter, the glass coverslips were dried with N_2_ then UV-O_3_ cleaned for 20 min. They were then immersed in a 1% solution of PEI in water (for glass-PEI) and nanopure water for bare glass. Coating was done overnight at RT with constant mixing at 60 rpm. After PEI immobilization, the glass slides were rinsed with copious amounts of nanopure water to remove the unbound PEI. Thereafter, the PEI-coated glass slides were washed with PBS. The bacteria suspension (OD = 0.1 in PBS) was incubated with the PEI-coated glass coverslips for 30 min before washing with PBS 2×, then with 0.85% NaCl for a third wash. PI/SYTO9 dissolved in 0.85% NaCl (1:2000 dilution) was added and incubated for 10 min before washing 3× with 0.85% NaCl. Finally, the stained bacteria were imaged using the Zeiss LSM510 META microscope.

## 3. Results and Discussion

### 3.1. PEI Immobilization Affects Its Ability to Change the Structure and Mechanical Stability of E. coli

First, we characterized *E. coli* immobilized on PEI-coated polycarbonate membranes (PC-PEI). PC membranes, with well-defined pore sizes, are mostly used for the physical entrapment of bacteria [27]. However, since it is relatively flat and is a highly negatively charged surface, we can readily coat it with PEI for bacterial attachment. Figure 1A shows an AFM topography image of *E. coli* immobilized on PC-PEI, where the bacteria retained its overall shape and size, with relatively smooth surface features (see Appendix A for additional images). This is consistent with our previous work, as well as with prior reports on AFM imaging of bacteria immobilized on PEI-coated substrates [16,17,25]. Furthermore, the overall structure and surface features of PEI-immobilized *E. coli* are very similar to the structural features of *E. coli* embedded within its biofilm matrix, also grown on the surface of PC membrane [26].

With the Quantitative Imaging (QI) mode of AFM, in addition to structural information, we were able to simultaneously measure the nanomechanical properties of immobilized *E. coli*. Figure 1B shows the histogram of the average apparent Young’s Modulus (YM) obtained from each of the 48 bacteria measured (see Appendix A for representative histograms of elasticity from individual bacteria). We highlight that the average apparent YM (7 ± 2 MPs) of these PC-PEI immobilized *E. coli* is in agreement with the apparent elasticity (also measured using AFM) of *E. coli* embedded within its biofilm matrix (i.e., 1–10 MPa) [26]. Moreover, the reported elasticity values for *E. coli* (AR3110) in this work falls within the same range as those obtained for other *E. coli* strains, as summarized by Tuson et al., where the Young’s moduli range from 0.12–12 MPa [28]. Furthermore, we tested the stability of PEI coating on the PC membrane and compared the structure and mechanical characteristics of *E. coli* in the first 60 min after immobilization (Figure 1A,B) vs. data obtained within the second hour after immobilization (Appendix A). We show that, within the first 120 min, the bacteria remain firmly bound on the PC-PEI membrane and no changes in both structure (Appendix A) and nanomechanical properties (Appendix A) were observed. In addition, the bacteria remained viable after PEI attachment. Figure 1C highlights that when immobilized *E. coli* is stained with a combination of SYTO9 and propidium iodide (PI), PI was unable to quench SYTO9 stained cells (Figure 1C), implying that the attachment to the PEI-coated surface does not compromise the integrity of the bacterial membrane. Taken together, our data suggest that while surface-bound hMW-bPEI can still bind and form a stable attachment with bacteria, it cannot induce changes in both bacterial structure and cellular mechanics. In addition, as long as PEI-surface interactions remain stable, the adhered bacteria retain their viability. Their shape, surface structure, as well as nanomechanical properties are generally similar to the *E. coli* embedded within their biofilm matrix [26].

We then tested whether this inability of hMW-bPEI to cause membrane disruption is brought about by its surface immobilization. Previous works have shown that for some antimicrobial peptides, their efficacy is greatly reduced upon surface immobilization [29]. These AMPs have linear structures and are usually specifically attached using just one functional group (mostly at one end of the polypeptide). For the branched PEI, however, the presence of multiple cationic charges along its branched structure implies that its immobilization on negatively charged surfaces would involve many of these cationic groups, and therefore would greatly affect its overall configuration compared to when the molecule is freely floating in the solution. Therefore, we pre-treated *E. coli* with unbound PEI for a few minutes (15 min) in PBS, before the bacteria were immobilized on the surface for AFM characterization. Figure 2A shows the topography of *E. coli* that was pre-treated with low concentrations (4 μg/mL) of PEI (see Appendix A for additional images). Unlike surface-bound PEI, it is evident that the binding of free PEI on the surface of *E. coli* was able to induce nanoscale undulations on the bacteria’s outer membrane. Although simple PEI deposition could potentially also change the surface topography of the bacteria, previous reports revealed that the hMW-bPEI can indeed cause major structural changes on the outer membrane of Gram-negative bacteria [11,30,31,32]. Helander et al. showed that major outer membrane undulations of vesicular nature can be observed by TEM imaging on PEI-treated *Salmonella typhimurium*, without changing its cytoplasmic membrane [11]. Krapf et al. have also shown, via AFM and TEM, that hMW-bPEI could induce major structural changes on the outer membrane of *Shewanella oneidensis* [31,32].

The structural changes observed after the exposure of *E. coli* to low concentrations of PEI did not, however, lead to changes in the nanomechanical property of the bacteria. Figure 2C shows that the distribution of the apparent Young’s modulus of PEI-decorated (low PEI) *E. coli* is statistically similar to the elasticity of non-treated *E. coli* (control). The fact that the bacteria’s cell mechanics remain unchanged even with significant structural changes on the outer membrane, shows that the bacterial envelope integrity of Gram-negative bacteria can withstand some outer membrane perturbations which are caused by membrane permeabilizers, such as PEI.

Next, we tested whether the mechanical stability of *E. coli* will hold at higher concentrations of PEI. We pre-treated the same concentration of *E. coli* with 100 μg/mL of PEI, in PBS for 15 min, before immobilizing the bacteria on the PC membrane. This time, the PEI-treated (high PEI) *E. coli* is no longer able to bind efficiently to PC-PEI, but is rather stable when immobilized on the bare PC membrane. This implies that at high PEI concentrations, the polymers have coated the surface of *E. coli* enough, thus it can no longer electrostatically bind to positively charged surfaces (PC-PEI), but bind more strongly to negatively charged surfaces (PC). Figure 2B shows that in the presence of high amount of PEI, the overall surface structure of *E. coli* has changed to a relatively smoother surface when compared to bacteria treated with low PEI concentrations, though some membrane undulations are still visible (see Appendix A for additional images). We believe that this is due to the almost complete coverage of the outer membrane in the presence of very high concentrations of PEI, leading to a more uniform surface feature, rather than the patches observed when the surface is only sparsely covered with the polymer. In addition to the observed structural changes, the apparent Young’s Modulus of *E. coli* decreased to 1.5 ± 0.5 MPa (from 7.5 ± 2.6 MPa for 4 μg/mL PEI), when pre-exposed to 100 μg/mL PEI (Figure 2C, high PEI, also see Appendix A for the effect of PEI immobilization on the nanomechanics of bare PC membranes). The dramatic change in cellular mechanics shows that given the right concentrations, outer membrane permeabilizers like hMW-bPEI can destabilize the integrity of bacterial cell envelope. This was also supported by the SYTO9/PI staining of bacteria exposed to unbound PEI, where higher PEI concentrations (100 ug/mL) lead to PI staining of the majority of *E. coli* cells (Appendix A). Furthermore, we show that unbound PEI could also induce structural and nanomechanical changes on *E. coli,* pre-immobilized on the surface of PC-PEI (Appendix A).

### 3.2. Pre-Treatment of E. coli with hMW-bPEI Does Affect Bacterial Growth, but Is Not Capable of Completely Killing the Bacteria Even at Very High Concentrations

Our data thus far has demonstrated that binding of PEI can induce structural and elasticity changes on *E. coli*, in a concentration-dependent manner. We next tested if these structural and nanomechanical changes have detrimental effects on *E. coli*, by monitoring its growth in liquid media after it was pre-treated with PEI. This is slightly different than the standard test used for determining the minimum inhibitory concentration (MIC) of an antimicrobial. Here, we first mix *E. coli* with PEI in a saline buffer (PBS) for 15 min before the bacteria-PEI solution was diluted in growth media, which does not contain the test reagent (i.e., PEI). These conditions mimic the duration of time the bacteria were exposed to free PEI during AFM imaging. Moreover, since PEI at 60 kDa forms cloudy solutions and precipitates in LB or M9 minimal media, they could significantly affect the absorbance reading used to monitor growth (data not shown). Figure 3A shows the growth curves of PEI-treated *E. coli* in M9 minimal media. It is evident from these growth curves that the 60 kDa branched PEI inhibits bacterial growth in a concentration dependent manner. However, as previously reported for other sizes of hMW-bPEI, this polymer is not completely bactericidal, even at very high concentrations (>1000 μg/mL). Detailed analysis of the growth curves (see Appendix A for details) indicates that while the maximal growth continues to decrease with increasing concentrations of PEI (Figure 3B), growth was still observed even at concentrations > 1000 ug/ml. Furthermore, the lag time and growth rate display a bimodal behavior—increasing initially before finally decreasing at even higher PEI concentrations (Figure 3C,D). It is also important to note that at lower PEI concentrations (i.e., 4 μg/mL), where we observed outer membrane structural changes but un-altered cell mechanics (Figure 2), we generally saw a similar behavior of growth between PEI-treated and non-treated *E. coli* (control). However, at higher concentrations (i.e., ~100 μg/mL), where we observed significant changes in both the outer membrane structure and cell mechanics, *E. coli* growth was inhibited. This implies that although not completely bactericidal, the PEI treatment of *E. coli*, at high concentrations, can disrupt the bacterial cell envelope enough to lower its cellular elasticity and inhibit its growth.

### 3.3. PEI-Decorated Polydopamine Nanoparticles Can Efficiently Target E. coli and Alter Cellular Mechanics

The presence of primary, secondary, and tertiary amino groups renders PEI with high cationic charge density, which is responsible for its stable attachment to negatively charged surfaces, such as PC-membranes and the surface of bacteria. The distribution of these cationic charges affects the overall architecture of the polymer once bound to polyanionic surfaces, changing their physicochemical characteristics. Above we have shown that the physical immobilization of PEI, on a flat surface, affects the way the polymers interact with and alter the properties of *E. coli* membranes. Next, we tested how the immobilization of PEI on a nanoparticle surface would affect its interaction with *E. coli*. Here, we used polydopamine nanoparticles (PDNP) as they can readily be functionalized with PEI using their primary amino groups [33]. Figure 4 confirms the successful functionalization of PDNP with PEI. The hydrodynamic size (${\mathrm{Size}}_{H}$) of the nanoparticles, as measured by DLS, increased by ~100 nm (Figure 4A). The hydrodynamic size of PEI alone is ~70 nm, which implies that PEI remained relatively flexible after nanoparticle immobilization. Moreover, the surface zeta potential of the nanoparticles changed after PEI attachment, from −22 to +30 mV in a 10 mM Bicine pH 8.5 buffer (Figure 4B). This dramatic shift from a negative to a positive surface zeta potential indicates that the surface of the highly negatively charged polydopamine has been completely covered with the highly cationic PEI. Both the increase in hydrodynamic size and the complete reversal of surface charge confirmed that PDNP was successfully coated with the polymer, forming PDNP-PEI particles.

The ability of these PDNP-PEI particles to bind to the surface of *E. coli* was then tested. Figure 4C shows that even at low concentrations of the particles (4 μg/mL, based on the dry weight of PDNP before PEI immobilization), several nanoparticles are already seen binding to the surface of *E. coli* (Appendix A for additional images). This demonstrates that PEI-decorated polydopamine nanoparticles are capable of targeting the bacterial membrane similar to what was observed when PDNP was decorated with an antimicrobial peptide [25]. Moreover, the topography image in Figure 4C shows that PDNP-PEI binding did not significantly alter the surface features of the bacteria (at least those that can be resolved using AFM in the QI mode). The outer membrane of *E. coli* remains largely smooth, similar to the control (Figure 1A), except in areas where the nanoparticles are bound. Despite these relatively unchanged surface features, the elasticity of *E. coli* slightly decreased after binding of PDNP-PEI (Figure 4E, low NP-PEI). The contribution of the nanoparticle elasticity was excluded from our analysis, as we were still able to select and analyze areas with no nanoparticles since the surface of *E. coli* is only sparsely covered with these particles (Appendix A). Moreover, we note that the binding of PDNP-PEI to the surface of *E. coli* is primarily driven by PEI, as we have previously shown that PDNP alone does not bind to the surface of bacteria and did not cause any observable changes in both topography and nanomechanics of *E. coli* [25].

When the concentration of PDNP-PEI was increased to 100 μg/mL, more nanoparticles were seen decorating the surface of the bacteria (Figure 4D). For this characterization, we used the bare PC-membrane, as the PDNP-PEI coated *E. coli* could no longer be immobilized on PC-PEI. This indicates that, similar to the incubation of *E. coli* with 100 μg/mL free PEI, the charge density of the bacteria surface has changed dramatically in the presence of high concentrations of PDNP-PEI. Despite the fact that it was decorated with more PDNP-PEI, still, no significant changes in the membrane structure were observed in areas not covered with nanoparticles. However, the apparent Young’s modulus of these bacteria are lower than those exposed to only 4 μg/mL PDNP-PEI (Figure 4D). For this analysis though, since the majority of the surface is now covered with the particles, the values that are shown in Figure 4D include an elasticity measurement done on the surface of PDNP-PEI attached to the bacteria (see Appendix A for the elasticity of PDNP-PEI). Since the apparent elasticity of *E. coli* decreased more in the presence of high concentrations of nanoparticles, immobilization of PEI on the surface of nanoparticles could still induce changes on bacterial cell mechanics in a concentration-dependent manner, similar to free PEI. However, the membrane disruptions are not as prominent as those imposed by free PEI, as seen from the elasticity values (compare Figure 2C and Figure 4E), SYTO9/PI staining (Appendix A), and also from the growth of *E. coli* pre-treated with increasing concentrations of PDNP-PEI (Appendix A). It remains to be seen, and our lab is actively looking into this, how the size of the nanoparticles would affect the ability of PDNP-PEI to induce membrane damage.

## 4. Conclusions

Surface immobilization of cationic polymer antimicrobials have been shown to reduce their effectivity. This is due to the fact that the overall structure of the polymer plays a major role in determining the nature of interactions with its target. Here, we have shown that this holds true for the membrane permeabilizer polycationic branched PEI. By monitoring both the structure and elasticity of the *E. coli* cell envelope, we have shown that the ability of PEI to interact with and change the physical characteristics of the bacterial cell envelope is largely dependent on the state of PEI: unbound, immobilized on a flat surface or attached to a nanoparticle.

## Figures and Tables

**Figure 1 microorganisms-09-02176-f001:**
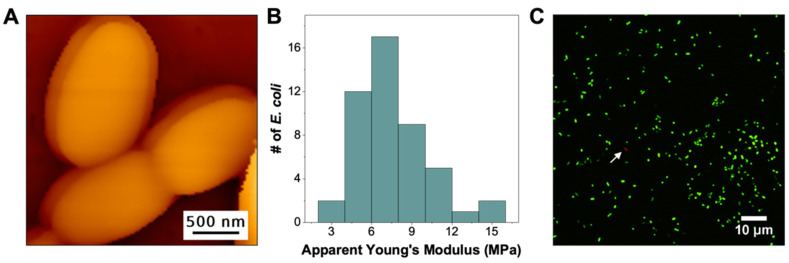
(**A**) Topography image of *E. coli* immobilized on the PEI-coated PC membrane. (**B**) Histogram of the average apparent Young’s Modulus of PC-PEI immobilized *E. coli* (*n* = 48). (**C**) Fluorescence image of PEI-immobilized *E. coli* stained with SYTO 9 (green) and propidium iodide (red) to test for membrane integrity. White arrow points to a PI-stained bacterium.

**Figure 2 microorganisms-09-02176-f002:**
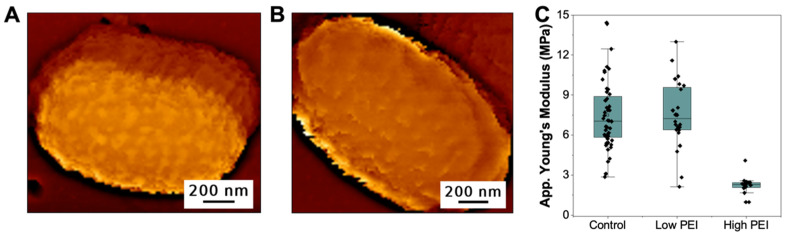
Topography images of *E. coli* pre-treated with (**A**) 4 μg/mL (*n* = 24) and (**B**) 100 μg/mL (*n* = 18) unbound PEI. (**C**) The distribution of the average apparent Young’s modulus of PEI-treated bacteria (low PEI = 4 μg/mL and high PEI = 100 μg/mL unbound PEI) relative to the untreated bacteria (control).

**Figure 3 microorganisms-09-02176-f003:**
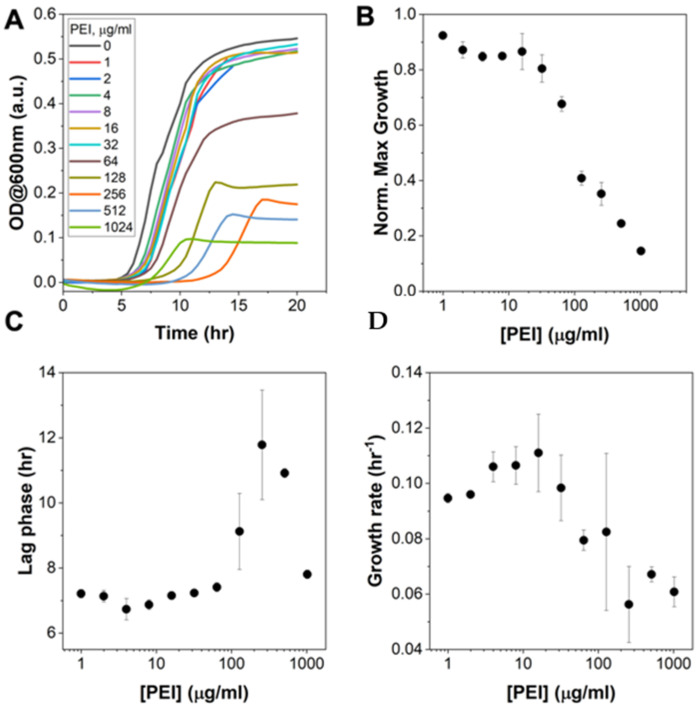
(**A**) Growth curves of *E. coli* in M9 minimal media after they were pre-treated with increasing concentrations of PEI (60 kDa) for 15 min. Using the Gompertz equation (see Appendix A) to fit the growth curves in (**A**), the maximal growth (**B**), lag time (**C**), and growth rate (**D**) were obtained as a function of PEI concentration.

**Figure 4 microorganisms-09-02176-f004:**
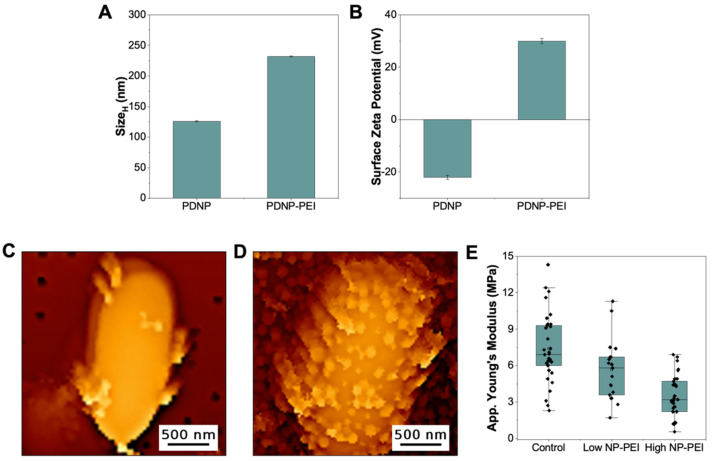
(**A**) Hydrodynamic diameter (${\mathrm{Size}}_{H}$
) and (**B**) surface zeta potential of polydopamine nanoparticles (PDNP) before and after functionalization with PEI (PDNP-PEI). Topography images of *E. coli* pre-treated with (**C**) 4 μg/mL (*n* = 20) and (**D**) 100 μg/mL (*n* = 31) PDNP-PEI. (**E**) The distribution of the average apparent Young’s modulus of PDNP-PEI treated bacteria (low NP-PEI = 4 μg/mL and high NP-PEI = 100 μg/mL PDNP-PEI) relative to the untreated bacteria (control).

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
