# Peer review of "Immobilization of Polyethyleneimine (PEI) on Flat Surfaces and Nanoparticles Affects Its Ability to Disrupt Bacterial Membranes"

_microorganisms, 2021, doi:10.3390/microorganisms9102176_

Round 1
Reviewer 1 Report
The ms by Andoy et al. describes the effects of surface immobilized, free and NP-immobilized PEI on bacterial cells. The authors used AFM imaging and nanoindentation, confocal microscopy and microbiology tests to decipher changes occurring on E. coli surface and viability when exposed to different PEI preparations. The work is of interest as it gives clues on how to use PEI for cell immobilization prior to AFM analysis and also on its use as antimicrobial agent. The methodology is clear, the ms is well written and conclusions are supported by the presented data. The topic is of broad audience interest. I recommend its publication in microorganisms after addressing the comments below.
1- For the mechanical analysis, more details should be given on the Hertz model used and the curve fitting, tip radius etc.
2- The Young modulus obtained here seems higher than other stiffnesses described for E. coli in the literature (below 1 MPa vs 3-12 MPa). This should be discussed in the ms and values should be compared to previous work.
3- I guess AFM measurements were achieved in liquid. Please precise the buffer, the temperature etc.
4- PC membranes were used for AFM whereas glass coverslips were used for confocal analysis. Why coverslips (+/- PEI) were not used for AFM?
5- Cell immobilized on PEI-PC membranes were analyzed after 1 and 2 hours (Fig. S3) but cells exposed to free PEI or PEI-coated NP were analyzed only after short exposition (15 min). Could the authors justify this exposure time?
6- How PDNP were weighed after synthesis? I guess NP are kept in liquid. Please precise the procedure.
7- Different MW exist for PEI. The reference used and the MW should be precised.
8- The effect of PEI-coated NP at different concentrations was determined but the analysis of uncoated NP is also necessary to conclude on the role of PEI.
9- Some sections lack references, i.e. first paragraph of results (lines 123-135), paragraph 3.1 (line 144-147) etc.
10- Line 130, the authors argue that PEI is non-bactericidal. References should be added to this sentence and it should be moderated. It is not totally true and may depend on the type, MW etc of PEI.
11- The results obtain here should be compared to the literature and the work on the effect of PEI on Shewanella oneidensis.
12- Fig. legend 1 and line 156: Young modulus obtained for 48 or 30 cells?
13- Line 164: the authors argue that not structural changes were observed after 2hr. But not images were presented (only Young modulus in S3). Please add images to S3 for 2 hr or remove the this part of the sentence.
14- Fluoresence with SYTO9 and PI was achieved only for cells immobilized on PEI-coated PC membrane after 1hour. I recommend to add fluorescence images of cells in all the tested conditions, i.e. after 2hr on PEI-PC membrane, after exposure to different concentrations of free PEI and after exposure to different concentrations of NP-PEI and bare NP. This would help to confirm and support many conclusions, e.g. line 220-222.
15- Most experiments were achieved in PBS. Phosphate from this buffer is negatively charged and may change the interaction between PEI and cells. Could you please comment on that?
16- The authors argue that cell topography changes after exposure to PEI. But there is no evidence that what is observed is a change in the cell envelope rather that simple PEI deposition on the surface. Please comment and moderate this conclusion.
17- When cells are exposed to 4ug/ml of PDNP-PEI, to which concentration of PEI does it correspond? This should be discussed in the ms.
18- Stronger effects and damages are observed with PDNP-PEI. Could it be due to higher local PEI concentration on the cell surface, i.e. NP concentrate locally PEI on the cell membrane which leads to membrane damages whereas with free PEI higher concentration is needed to induce damages?
Typo/vocabulary:
- When liquid contains cells, “solution” should be replaced by “suspension”.
Reviewer 2 Report
Andoy et al tackle in their manuscript the influence that a supporting substrate, either flat or colloidal, has on the bactericidal activity of positive polyelectrolyte poly-ethyleneimine (PEI). For such a purpose, authors have performed atomic force microscopy measurements (simultaneous imaging and force, as provided by QI mode) on bacteria electrostatically attached to polycarbonate membranes, through a PEI coating. Attachment was conducted under three different conditions: untreated bacteria, PEI-coated bacteria (under 2 PEI concentrations) and, finally, bacteria which had been exposed to dopamine-PEI nanoparticles (also in two different concentrations) prior to their attachment. By means of topography imaging, authors could detect structural changes caused on the outer membrane, while simultaneous force measurements enabled characterizing potential variations on the elastic response derived from those structural changes. Based on their results, authors conclude that freely standing branched PEI chains, above a certain concentration value, act more effectively on the disruption of bacterial membrane integrity.
Attending to my personal opinion, the study is quite optimally explained and reproducible, both measuring approaches as well as the data analysis performed are suitable for answering the hypothesis posed (with special mention for the beautiful AFM images obtained), and the selected references throughout the text are well employed. However, after a thorough reading, I find that there are a few open questions that would need to be answered before the manuscript is accepted for publication in Microorganisms journal.
The main comments/suggestions would be the following (that in some cases might involve performance of some measurements):
- Authors explain the influence of exposing bacteria to free (soluble) PEI before attachment but, have they performed measurements on already attached bacteria (as those in section 3.1)? How would that affect bacterial membrane integrity and attachment?
- I somehow missed control measurements of the elastic modulus of flat PEI layers (on glass) and that of dopa-PEI particles. These control values could be then employed to discard that the drop observed for PEI-systems in high concentrations is due to the presence of this coating and not because of membrane disruption.
- In a similar line, is there any reason not to perform SYTO9/propidium iodide treatment (or at least not to show it) upon treatment with free PEI and the dopamine-PEI nanoparticles? I guess such a study would determine if bacterial membrane gets affected.
Additional (minor) things are:
- References should appear before punctuation.
- Page 2 Line 67 (P2L67): I would slightly reformulate this sentence for a better understanding. Suggestion: “It has been suggested that only polymers that detach from the surface might cause denaturation of bacteria”.
- The nanoparticle synthesis here described should be linked to the characterization that is included in Figure 4. Alternatively, those plots (4a and 4b) could be moved to the supporting information, so the rest of the figures (4c-e) don’t lose their strength. I also found that most of the text from the initial paragraphs from section 3.3 could be part of Materials and Methods instead.
- What is the reason of the pH variations between 8.5 and 7.5 for Bicine buffer during the particle preparation?
- replace the symbol for µL (instead of uL)
- The description of the bacterial growth analysis should include a more detailed explanation of the fitting applied (Gompertz eq) and the values derived from it.
- I was surprised about the usage of an alternative software for AFM data analysis? Which are the main benefits/differences with JPK DP software? (just personal interest)
- The introduction to results and discussion is pretty redundant (and so is that for section 3.3) regarding the information already provided in the main introduction.

Reviewer 3 Report
The authors describe the synthesis of PEI functionalized polydopamine NPs and their interaction with E. coli. In particular, the effect of the PEI immobilization on its antimicrobial abilities. Such ability is reduced depending on the state of PEI: free, immobilized on a PC membrane or on Polydopamine NP surface.
I find this work well presented and complete, I suggest its publication after minor revision
Author Response
Reviewer 3 did not raise any concerns.
Round 2
Reviewer 1 Report
The authors addressed my comments
Reviewer 2 Report
After I went thoroughly through the changes applied in the manuscript by Andoy et al., I can conclude that the overall quality has been definitely improved.Therefore, it now deserves acceptance for publication in Microorganisms journal.